# Soluble Guanylate Cyclase β1 Subunit Represses Human Glioblastoma Growth

**DOI:** 10.3390/cancers15051567

**Published:** 2023-03-02

**Authors:** Haijie Xiao, Haifeng Zhu, Oliver Bögler, Fabiola Zakia Mónica, Alexander Y. Kots, Ferid Murad, Ka Bian

**Affiliations:** 1Department of Biochemistry and Molecular Medicine, The George Washington University, 2300 I Street NW, Washington, DC 20037, USA; 2The Brown Foundation Institute of Molecular Medicine for the Prevention of Human Diseases (IMM), The University of Texas Health Science Center at Houston, 7000 Fannin Street, Houston, TX 77030, USA; 3Department of Genomic Medicine, The University of Texas MD Anderson Cancer Center, 1515 Holcombe Blvd., Houston, TX 77030, USA; 4Brain Tumor Center, The University of Texas MD Anderson Cancer Center, 1515 Holcombe Blvd., Houston, TX 77030, USA; 5The National Cancer Institute, NIH, 9000 Rockville Pike, Bethesda, MD 20892, USA; 6Department of Pharmacology, Faculty of Medical Sciences, State University of Campinas (UNICAMP), Campinas, Sao Paolo 13083, Brazil; 7Veteran Affairs Palo Alto Health Care System, Department of Veteran Affairs, Palo Alto, CA 94304, USA

**Keywords:** glioblastoma, sGCβ1, nucleus, p53, CDK6, integrin α6, G0 arrest

## Abstract

**Simple Summary:**

A marked reduction in soluble guanylyl cyclase β1 (sGCβ1) transcript is characteristic for human glioma specimens. Restoring the expression of sGCβ1 inhibited the aggressive course of glioblastoma in an orthotopic xenograft mouse model. The present study is the first to reveal that sGCβ1 migrated into the nucleus and interacted with the promoter of the *TP53* gene. sGCβ1 overexpression impacted signaling in glioblastoma multiforme, including the promotion of nuclear accumulation of p53, a marked reduction in cyclin-dependent kinase 6 (CDK6), and a significant decrease in integrin α6. Antitumor effect of sGCβ1 was not associated with enzymatic activity of sGC.

**Abstract:**

Malignant glioma is the most common and deadly brain tumor. A marked reduction in the levels of sGC (soluble guanylyl cyclase) transcript in the human glioma specimens has been revealed in our previous studies. In the present study, restoring the expression of sGCβ1 alone repressed the aggressive course of glioma. The antitumor effect of sGCβ1 was not associated with enzymatic activity of sGC since overexpression of sGCβ1 alone did not influence the level of cyclic GMP. Additionally, sGCβ1-induced inhibition of the growth of glioma cells was not influenced by treatment with sGC stimulators or inhibitors. The present study is the first to reveal that sGCβ1 migrated into the nucleus and interacted with the promoter of the *TP53* gene. Transcriptional responses induced by sGCβ1 caused the G0 cell cycle arrest of glioblastoma cells and inhibition of tumor aggressiveness. sGCβ1 overexpression impacted signaling in glioblastoma multiforme, including the promotion of nuclear accumulation of p53, a marked reduction in CDK6, and a significant decrease in integrin α6. These anticancer targets of sGCβ1 may represent clinically important regulatory pathways that contribute to the development of a therapeutic strategy for cancer treatment.

## 1. Introduction

Glioblastoma is an aggressive brain cancer known to be resistant to treatment. Over 13,000 Americans were estimated to have been diagnosed with glioblastoma in 2022, and over 10,000 people in the United States are diagnosed with glioblastoma on a yearly basis. Glioblastoma patients are characterized by 6.8% five-year survival rate, and the average survival of these patients is estimated to be only 8 months. The rate of survival and overall mortality due to glioblastoma have been essentially unchanged for many years [1]. It was suggested that veterans who were deployed to Iraq and Afghanistan developed glioblastoma at a higher rate [2,3,4,5]. Thus, a new therapeutic strategy for glioblastoma is necessary.

Our previous study analyzed the changes in the signaling molecules of the nitric oxide (NO)/soluble guanylyl cyclase (sGC)/cyclic guanosine monophosphate (cGMP) pathway in the human glioma tissues and cell lines and compared the levels of these molecules with normal controls. We demonstrated that the expression of sGC is significantly lower in glioma preparations. The restoration of sGC expression by genetic manipulations or elevation of the level of intracellular cGMP using pharmacological agents in glioblastoma cells was shown to significantly suppress the growth of tumor cells. Orthotropic implantation of human glioblastoma cells transfected with a constitutively active mutant form of sGC (sGCα1β1^cys105^) in athymic mice was demonstrated to induce a four-fold increase in survival time compared with that in the control group [6].

sGC functions as a heterodimer composed of the α and β subunits, and α1/β1 sGC is the most abundant heterodimer [7]. The α1 and β1 subunit genes are located in the same chromosome 4 in humans and are encoded by separate genes [8]. The heterodimer is required for enzyme function. However, the levels of the α1 and β1 subunits of sGC can be independently regulated in most human tissues [9]. sGC is gaining a rapidly growing interest as a therapeutic target. The first-in-class sGC stimulator riociguat was approved for pulmonary hypertension in 2013, and another stimulator, vericiguat, was recently approved in the USA for patients with heart failure. These sGC stimulators enhance sGC activity independently from NO [10,11].

Our previous analysis has shown that higher levels of sGCβ1 in cancer tissues are correlated with greater survival probability of patients compared with that in patients with lower levels of sGCβ1 [12]. The present study reported that restoration of sGCβ1 expression in sGC-deficient human glioblastoma cells blocked the aggressive course of malignant tumors independently of cGMP. In contrast to sGCα1, the sGCβ1 subunit migrated into the nucleus and bound to the chromatin complex. The data of the present study will help advance our understanding of the role of sGCβ1 in glioma proliferation.

## 2. Materials and Methods

### 2.1. Plasmid Construction and Cell Culture

The protocol for the generation of the stable clones of human glioblastoma U87 cells by transfection was described in a previous study [6]. Briefly, full-length sGCβ1 and sGCβ1^Cys105^ were cloned into the pcDNA3.1D/V5-His TOPO vector (Invitrogen) according to the manufacturer’s instructions. For sGCβ1 knockdown, BE2 cells were transfected with nonsilencing control shRNA or sGCβ1 shRNA (Origene). The structures of all plasmids used in the present study were confirmed by sequencing. The changes in the expression of sGCβ1 were confirmed by Western blot analysis prior to the experiments [6].

Glioblastoma U87 and neuroblastoma BE2 cells were obtained from the American Type Culture Collection (Manassas, VA, USA) and maintained at 37 °C under a humidified atmosphere containing 5% CO_2_. U87 cells were grown in Dulbecco’s modified Eagle’s medium, and BE2 cells were grown in Eagle’s minimal essential medium. Medium was supplemented with 10% fetal bovine serum (FBS, HyClone, Logan, UT, USA) and 1% penicillin/streptomycin mixture.

### 2.2. Cell Viability Assay

Cell viability and proliferation were measured by the MTT (3-(4,5-dimethylthiazol-2-yl)-2,5-diphenyl-2H-tetrazolium bromide) method. The cells with sGCβ1 overexpression or knockdown were seeded at 20,000 cells/mL in a 96-well plate for at least 24 h and incubated further as described in the text. Then, MTT was added to the culture at a final concentration of 0.5 mg/mL, and the samples were incubated for 3 h. Then, medium was aspirated, and 100 µL/well of anhydrous isopropanol containing 40 mM hydrochloric acid was added. Optical density was read at 570 nm to evaluate the proliferation of the cells. For pharmacological treatments, the cells were treated with DMSO (dimethyl sulfoxide) (0.1%; vehicle control), ODQ (1H-[1,2,4]oxadiazolo[4,3-a]quinoxalin-1-one; 10 µM), Bay41-2272 (3-(4-amino-5-cyclopropylpyrimidine-2-yl)-1-(2-fluorobenzyl)-1*H*-pyrazolo[3,4-*b*]pyridine; 1 µM), or YC-1 (3-(5′-hydroxymethyl-2′-furyl)-1-benzylindazole; 10 µM) for 24 h prior to the MTT assay.

### 2.3. Colony Formation Assay

Traditional soft agar assay for colony formation was used with some modifications. U87 glioblastoma cells (3000 cells) were seeded in an agar–agarose semisolid gel system in a 30 mm cell culture dish covered by culture medium containing FBS, and the medium was replaced every three days. For pharmacological treatments, the cells were treated with DMSO (0.1%; vehicle control), ODQ (10 µM), Bay41-2272 (1 µM), or YC-1 (10 µM). The medium containing the reagents was replaced every three days. After 21 days, the colonies were stained with 0.005% crystal violet for 1 h, washed with PBS, and imaged. The colonies were counted, and the size of the colonies was calculated.

### 2.4. Orthotopic Xenograft Models

The animal protocol was approved by the Institutional Animal Care and Use Committee of The University of Texas MD Anderson Cancer Center (protocol no. 10-07-12131), and all experiments were performed according to the National Institutes of Health guidelines.

A total of 14 female mice (8–10-week-old; nu/nu athymic; Charles River Laboratories) were used in the experiments. Human glioblastoma cell lines with or without stable transfection (at a concentration of 1 × 10^6^ cells/5 µL) were resuspended in PBS and injected into the right frontal lobe brain of nude mice by using a guide-screw system as described previously [6]. The animals were anesthetized with xylazine-ketamine (10 mg/kg xylazine and 100 mg/kg ketamine). The animals were euthanized when they became moribund due to tumor progression. The brain was then removed for histological and molecular analyses.

### 2.5. Assay of cGMP in Intact Cells

To assay the accumulation of cGMP in the tumor cells, the cells were preincubated in Dulbecco’s PBS containing 1 mM 3-isobutyl-1-methylxanthine for 10 min. The medium was aspirated, and 50 mM sodium acetate (pH 4.0; 0.3 mL per well) was added to extract cGMP by rapid freezing of the plates at −80 °C. The accumulation of cGMP was measured by enzyme-linked immunosorbent assay (ELISA) as described previously [13].

### 2.6. Quantitative Real-Time Polymerase Chain Reaction (qRT-PCR)

Total RNA was isolated using Trizol reagent (Thermo Fisher Scientific, Waltham, MA, USA) according to the manufacturer’s instructions. Complementary DNA was synthesized by using an iScript™ reverse transcription supermix kit (cat. no. 170-8840; Bio-Rad, Hercules, CA, USA) following the manufacturer’s protocol. qRT-PCR was performed using a standard protocol in a total reaction volume of 25 μL as described previously [14] and as follows: 2 min at 95 °C, followed by 40 cycles of 15 s each at 95 °C and 30 s at 60 °C. The data of qRT-PCR were normalized to the level of β-actin. The PCR primers used in the present study are listed in Appendix A.

### 2.7. Western Blot Analysis

The cells were harvested and lysed by sonication in ice-cold RIPA buffer containing a protease inhibitor cocktail. Isolation of the protein fractions from the cytoplasm and soluble nuclear and chromatin fractions is described in the Appendix A. Equal amounts of the protein (50 or 100 µg/lane) were separated by 4–15% SDS-PAGE. Separated proteins were transferred to a polyvinilydene difluoride (PVDF) membrane, which was blocked with 5% nonfat dry milk in TBS-T (20 mM Tris-HCl, pH 7.6, 130 mM NaCl, and 0.1% Tween 20) and incubated at 4 °C overnight with specific primary antibodies. Secondary horseradish peroxidase-conjugated antibodies (Sigma-Aldrich, St. Louis, MO, USA) were used at 1:3000–1:10,000 dilutions, and the protein bands were visualized by enhanced chemiluminescence (ECL Plus; Amersham Biosciences, Buckinghamshire, UK). The intensity of the bands was quantified using ImageJ software (NIH). All antibodies used in the present study are listed in Appendix A.

### 2.8. Flow Cytometry

The cells were collected by trypsinization and fixed with 70% ethanol overnight at 4 °C. After fixation, the cells were centrifuged and stained in 1 mL of propidium iodide solution (0.05% NP-40, 50 ng/mL propidium iodide, and 10 μg/mL RNase A). Labeled cells were analyzed using a BD flow cytometer and FlowJo software, Mississauga, ON, Canada.

Double staining of U87 cells was performed according to Li et al. [15]. Briefly, 1 × 10^6^ PBS-washed cells were incubated in 0.5 mL of nucleic acid-staining solution (NASS) containing 0.02% saponin and 10 µg/mL 7AAD (7-aminoactinomycin D) for 20 min at room temperature protected from light. After the addition of 1 mL of PBS to the samples, the cells were centrifuged at 250 g for 5 min. Cell pellet was resuspended in 0.5 mL of NASS containing 10 µg/mL actinomycin D. The mixture was incubated on ice for 5 min protected from light. Pyronin Y was added to the cells to a final concentration of 1 μg/mL, and the suspension was vortexed. The cells were incubated on ice protected from light for at least 10 min before the data were acquired using a flow cytometer.

### 2.9. Confocal Microscopy

Nuclear localization of sGCβ1 was determined by indirect immunofluorescence. In brief, the cells were grown on sterile glass coverslips, fixed in 4% paraformaldehyde, permeabilized using 0.1% Triton X-100, and blocked with 10% normal goat serum in PBS. The cells were incubated with primary antibodies, washed three times in PBS, and incubated with goat antimouse or goat antirabbit secondary antibodies conjugated with fluorescein isothiocyanate (FITC; green; Molecular Probes, Eugene, OR, USA). The nuclei were stained with the blue DNA dye 4′,6-diamidino-2-phenylindole (DAPI; Molecular Probes). The images were acquired by an Olympus FV300 laser-scanning confocal microscope using sequential laser excitation to minimize fluorescence emission bleed-through.

### 2.10. Chromatin Immunoprecipitation (ChIP) Assay

ChIP assay was performed by using a SimpleChIP enzymatic chromatin immunoprecipitation kit (cat. no. 9003; Cell Signaling, Danvers, MA, USA) according to the manufacturer’s instructions. Briefly, 4 × 10^7^ cells were fixed with 1% formaldehyde for 10 min, and the reaction was quenched with 0.125 M glycine. The cells were washed with PBS, and the nuclei were isolated by incubating the cells in buffer A for 10 min on ice. Micrococcal nuclease was then used to digest the chromatin for 20 min at 37 °C. The samples were sonicated by a Qsonica sonicator at 40% amplitude for 20 s three times. An anti-sGCβ1 antibody (Sigma; cat. no. G4405; 3.5 µg) or control IgG was added to the chromatin samples and incubated overnight with rotation. Then, magnetic beads were added to the chromatin samples and incubated at 4 °C for 2 h with rotation. After three washes with a low-salt buffer and one wash with a high-salt buffer, DNA was eluted from the beads with elution buffer and purified using a column. qRT-PCR was performed to detect sGCβ1 binding to the chromatin regions.

### 2.11. Dual Luciferase Assay

Luciferase assay was used to evaluate the activity of the *TP53* promoter. U87 cells were grown to 50% confluence in 24-well plates and transfected by using Lipofectamine LTX reagent (Invitrogen, Waltham, MA, USA) according to the manufacturer’s instructions. The cells were transfected with 0.1 μg of the pGL3-TP53 (or a corresponding deletion mutation) vector in combination with 0.2 µg of the sGCβ1 overexpression plasmid (pCDNA3.1 empty vector was used as a control) and pRL-SV40 (Promega, Madison, WI, USA). Twenty-four hours after the transfection, the luciferase activity was determined by using a dual luciferase assay kit (Promega) according to the manufacturers’ protocol, and the signal was acquired using a Biotek Synergy H1 microplate reader (Agilent Technologies, Santa Clara, CA, USA).

### 2.12. Statistical Analysis

The results are expressed as the mean ± S.E.M. One-way analysis of variance (ANOVA) was used for multiple comparisons, and the results were Bonferroni-adjusted. Significance of the differences between the treatment groups was assessed by Student’s *t* test versus the control groups using Welch correction as appropriate; the *p*-values of less than 0.05 were considered statistically significant. The *n* values indicate the numbers of animals or independent biological replicates used in the experiments. The Kaplan-Meier survival curves and the mean survival values were compared using the log-rank test (SigmaPlot version 12.5 software) and were Bonferroni-adjusted for multiple comparisons.

## 3. Results

### 3.1. sGCβ1 Overexpression Represses the Growth of Human Glioblastoma

We have previously demonstrated that the expression levels of α1 and β1 subunits of sGC are significantly lower in human glioma preparations [6]. We analyzed the data of SAGE (serial analysis of gene expression; GEO databases GSE15309; *n* = 327) based on the mRNA sequencing output and demonstrated a statistically significant reduction in sGCβ1 transcript levels in human glioma specimens compared with that in normal adjacent tissues (Figure 1a) [16]. To examine the effect of genetically restored sGCβ1 expression on the growth of glioma cells, we generated two stable clones of U87 human glioblastoma cells overexpressing unmodified sGCβ1 or mutant sGCβ1^Cys105^. The sGCβ1^Cys105^ mutant was created by substituting Cys for His at position 105 [17]. The sGCα1β1^Cys105^ heterodimer is constitutively active and is not stimulated by nitric oxide [17]. The proliferation of U87 cells was significantly inhibited by overexpression of either sGCβ1 or sGCβ1^Cys105^ (Figure 1b). Our previous study has shown that the delivery of sGCα1 alone failed to suppress the proliferation of human glioma cells [6]. The growth of the cells within a three-dimensional (3D) support system is known to simulate a natural microenvironment for the proliferation, morphology, signaling, and responses to therapeutic agents [18]. Thus, we used a colony formation assay to evaluate the influence of sGCβ1 on the growth of glioblastoma cells. As shown in Figure 1c, the expression of sGCβ1 or sGCβ1^Cys105^ decreased the number and size of the colonies of glioblastoma cells. The colonies formed by the stable clones overexpressing sGCα1 were similar to the colonies formed by control cells [6].

Next, we performed an in vivo study by orthotopic xenotransplantation of U87 cells with or without prior transfection of sGCβ1^Cys105^. As shown in Figure 1d, the animals inoculated with sGCβ1^Cys105^-transfected cells had significantly extended survival time; the longest survival time of the animals inoculated with sGCβ1 subunit-transfected cells was over 125 days (four-fold increase over the control group). The average survival time of mice inoculated with the sGCβ1^Cys105^-expressing cells was increased from 29 ± 2 days to 69 ± 11 days (Figure 1e). However, the intracranial xenograft of sGCαl-transfected cells prolonged the average survival time from 31 days to only 40 days [6], indicating that sGCβ1 was significantly more potent in suppression of tumor growth in vivo.

The antiproliferative effect of sGCβ1 suggested to test whether silencing of sGCβ1 enhances the growth of other tumor cells. BE2 human neuroblastoma cells normally express both α1 and β1 subunits of sGC similar to the expression pattern detected in a normal human cortex [6,19,20,21]. Thus, BE2 cells were selected to study the effect of gene silencing. sGCβ1 was consistently silenced after the transfection with short hairpin RNA (shRNA). As shown in Figure 1f, a reduction in sGCβ1 expression resulted in an increase in the proliferation of BE2 neuroblastoma cells.

### 3.2. The Growth Repression by sGCβ1 Overexpression Is cGMP-Independent

To determine if the repression of the proliferation is due to cGMP overproduction, we examined the cGMP levels in the cells with sGCβ1 overexpression. sGCβ1 or sGCβ1^Cys105^ overexpression in U87 cells did not significantly change the cGMP levels compared with that in untransfected or control vector-transfected glioblastoma cells (Figure 2a). To assess possible involvement of enzymatic activity of sGC in the effects, we examined the influence of sGC inhibitors or activators in sGCβ1-overexpressing cells. The proliferation rate of stable clones overexpressing sGCβ1 or sGCβ1^Cys105^ was not influenced by the presence of the sGC inhibitor ODQ or by activators Bay41-2272 and YC-1 (Figure 2b). The colony formation by the cells overexpressing sGCβ1 or sGCβ1^Cys105^ was not changed by treatment with ODQ, Bay41-2272, or YC-1 (Appendix A). Thus, these results indicated that the growth repression by sGCβ1 was not associated with cGMP production.

### 3.3. sGCβ1 Overexpression Induces G0 Phase Arrest of Human Glioblastoma Cells

The cell cycle phases of sGCβ1-overexpressing cells were analyzed by flow cytometry after the cells were stained with propidium iodide. As shown in Figure 3a, the G0/G1 phase was prolonged in sGCβ1-overexpressing cells because the number of the cells in the G0/G1 phase was higher than that in the control cells by 14%. To distinguish between the G0 and G1 phases, the cells were double stained with 7AAD and Pyronin Y. As shown in Figure 3b, sGCβ1-overexpressing cells had a significantly higher population of the cells in the G0 phase (17.6% in sGCβ1-overexpressing cells versus 4.8% in control U87 cells). No significant changes in the sub-G1 phase population were detected in sGCβ1-overexpressing cells after staining with propidium iodide or 7AAD, suggesting that sGCβ1 overexpression had no influence on apoptosis.

### 3.4. sGCβ1 Is Localized in the Nucleus in Human Glioblastoma Cells

The α1 and β1 subunits of sGC have variable distribution in various tissues and can be regulated independently under certain conditions. Moreover, intracellular sGCα1 and β1 can localize to various intracellular compartments [22,23,24]. As shown in Figure 4a, a significant portion of sGCβ1 was detected in the nuclear fraction of U87 cells overexpressing sGCβ1 or coexpressing the α1 and β1 subunits. Subcellular sGCβ1 localization patterns were similar in U87 cells cotransfected with or without sGCα1 (Figure 4a), suggesting that the formation of the functional heterodimer is not required for nuclear localization of sGCβ1. In contrast, sGCα1 was not detected in the nucleus even in sGCα1-overexpressing U87 cells (Figure 4b). Confocal fluorescence imaging analysis confirmed nuclear localization of sGCβ1 (Figure 4c).

### 3.5. A Link between sGCβ1 and p53

p53 is a well-known tumor suppressor that primarily alters the expression of numerous genes involved in cell cycle arrest, apoptosis, stem cell differentiation, and cellular senescence. The p53 pathway is frequently deregulated in glioblastoma, and this deregulation is correlated with a more invasive, more proliferative, and more stem-like phenotype [25]. The data of qRT-PCR showed a significant upregulation of *TP53* gene expression after sGCβ1 overexpression (Figure 5a). The level of p53 protein was increased by approximately 50% after overexpression of sGCβ1 (Figure 5b). Wild type p53 localizes in the cytoplasm and nucleus of human primary glioblastomas [26]. Wild type p53 and naturally occurring p53 mutants migrate into the nucleus, and nuclear localization of p53 plays essential roles in tumorigenesis and malignant transformation. Cytoplasmic p53 associates with microtubular cytoskeleton to localize to the mitochondria during nontranscriptional apoptotic response [27,28,29]. The expression of p53 can be detected in the cytoplasmic and nuclear fractions of human glioblastoma U87 cells (Figure 5b,c), which express wild type p53. Boosting the expression of sGCβ1 markedly elevated nuclear accumulation of p53 and had a less pronounced effect on cytosolic p53, suggesting that sGCβ1 overexpression may enhance the interaction of p53 with the nuclear components (Figure 5b,c).

To further explore the role of sGCβ1 in the nucleus, we examined possible transcription activity of sGCβ1 targeting the *TP53* promoter by ChIP. The binding site of sGCβ1 to the *TP53* gene was enriched in the region 1039 bp downstream of the transcription start site (TSS), with the MACS (model-based analysis of ChIP-sequencing) *p*-value of 184.44 (Appendix A). The data of ChIP obtained using a specifically designed set of primers (Figure 5d) clearly indicated that the binding of sGCβ1 was markedly enriched at approximately 1 kb downstream of the TSS of the *TP53* promoter (Figure 5d). To confirm the binding of sGCβ1 to the *TP53* promoter region, the promoter of the *TP53* gene was cloned into the pGL3 vector, and site-directed mutagenesis was performed (Figure 5e). The plasmids containing the *TP53* promoter with or without the deletions in combination with sGCβ1-overexpressing plasmid and a control reporter vector pRL-SV40 were transfected into U87 cells. As shown in Figure 5e, the results of a dual luciferase assay indicated that sGCβ1 overexpression activated the *TP53* promoter. When the putative sGCβ1-binding site was deleted, the activity of the *TP53* promoter was not influenced by sGCβ1 overexpression. Thus, we confirmed that sGCβ1 binds to and activates the *TP53* promoter.

### 3.6. sGCβ1 Overexpression Represses Glioblastoma Multiforme Signaling

Two highly conserved p53-responsive elements in the p21 promoter directly bind p53 to activate p21 transcription [30]. p53-mediated apoptosis is preceded by an elevation in the levels of p21 [31]. To verify the effect of sGCβ1 on the p53 signaling pathway, we examined the status of p21 expression. As demonstrated in Figure 6a, mRNA levels of p21 were increased by approximately 21%, and the protein levels of p21 were increased by approximately 48% in sGCβ1-overexpressing cells compared with those in control human glioblastoma cells (Figure 6b,c).

The retinoblastoma tumor suppressor protein (Rb) and p53 pathways appear among the most frequently mutated pathways in malignant glioma. The retinoblastoma family of proteins, including Rb and p107, undergoes cell cycle-dependent phosphorylation during the G1 to S phase transition, and p130 is phosphorylated during the G0 and early G1 phases of the cell cycle [32]. We sought to determine the role of sGCβ1 in the regulation of the phosphorylation levels of p130, p107, and pRb. However, sGCβ1 overexpression did not influence the phosphorylation levels of the members of the retinoblastoma family (Appendix A).

CDKs are the key regulators of the retinoblastoma family to promote cell cycle progression, which is crucial for pathological process of cancer. Glioblastoma is characterized by a high frequency of CDK4/CDK6 pathway dysregulation [33]. Pharmacological blockage of CDK4/6 has been recently investigated in clinic and demonstrated promising activity in patients with breast and other cancers [34]. The data of qRT-PCR and Western blotting obtained in the present study demonstrated that sGCβ1 overexpression markedly inhibited the expression of CDK6 and induced a trend of downregulation of CDK4 (Figure 6d–f).

Glioblastoma displays with remarkable cellular heterogeneity. Glioblastoma stem-like cells are the key players among various cellular elements [35,36]. Integrin alpha 6 (ITGA6) has received considerable attention due to its role in the regulation of glioblastoma stem-like cells [37,38]. As shown in Figure 6g–i, mRNA and protein levels of ITGA6 were significantly downregulated in sGCβ1-overexpressing cells.

## 4. Discussion

The data of the present study demonstrated that sGCβ1 expression was significantly reduced in human glioma preparations (Figure 1a). The results of the proliferation and 3D colony formation assays demonstrated that restoration of sGCβ1 expression played a critical role in the blockade of the growth of human glioblastoma cells (Figure 1). The growth-repressing effect of sGCβ1 was cGMP-independent since we did not detect significant changes in the inhibitory effect of sGCβ1 after treatment with the activators or an inhibitor of sGC activity (Figure 2). Orthotopic xenograftment with an sGCβ1^Cys105^-expressing stable clone of glioblastoma cells in athymic mice increased the maximal survival time of the animals four-fold compared with that in the vector control group.

The role of the NO and cGMP signaling pathway in biological properties of the tumors has been actively investigated during the past 30 years. However, this pathway may be beneficial or detrimental for cancer. Several reasons for this ambiguity can be considered. NO participates in normal signaling (e.g., vasodilation and neurotransmission); however, NO has cytotoxic or proapoptotic effects when produced at high concentrations by inducible nitric oxide synthase (iNOS or NOS-2). In addition, the levels of the cGMP-dependent (the NO/sGC/cGMP pathway) and cGMP-independent (the NO redox pathway) components vary between various tissues and cell types. Frequent deregulation of sGC expression at the levels of transcription [39], splicing [40,41], mRNA stability [39], and protein stability [42] have been investigated by us and recently reviewed [43].

Solid tumors include two compartments, the parenchyma (neoplastic cells) and stroma (nonmalignant supporting tissues including connective tissue, blood vessels, and inflammatory cells), and biological properties and signaling pathways influenced by NO are different in these compartments. Thus, specific features of the NO/sGC/cGMP signaling pathway should be further characterized in the tumor and surrounding tissues [6,44]. Our previous study provided evidence for two possible roles of NO/cGMP signaling in malignant tumors. First, NOS-2 expression and NO overproduction contribute to the formation of an inflammatory cancer microenvironment, which promotes tumor cell proliferation. Second, a deficiency in sGC/cGMP signaling diminishes the role of these molecules as antagonists of cancer cell growth [6,44]. The present study revealed that sGCβ1 alone can migrate into the nucleus, thus impacting malignant cellular signaling to change the course of tumor growth. Consistent with this finding, the studies of sGCβ1 in rat brain astrocytes and glioma cells demonstrated that sGCβ1 is localized in the nucleus and is associated with the chromosomes during mitosis, regulating chromatin condensation and cell cycle progression in a cGMP-independent manner [45].

The p53 pathway is frequently inactivated in glioblastomas (for review, see [46] and references therein). Genome-wide system analyses based on transcriptome profiles have stratified gliomas into four molecular signatures: proneural, neural, classic, and mesenchymal [47]. The inactivation of the p53 pathway is common for glioblastomas of all four subgroups. The restoration of the expression of sGCβ1 in human glioblastoma cells significantly promoted p53 expression at mRNA and protein levels and correlated with the growth-suppressing effect of this sGC subunit (Figure 4 and Figure 5). The effect of sGCβ1 over expression on the level of *TP53* mRNA was somewhat lower than the effect on the level of the p53 protein, which is a relatively common phenomenon for p53 [48]. Numerous studies have investigated the roles of cell cycle arrest and apoptosis in tumor suppression by p53 because these processes are the most evident antitumor mechanisms [49]. The human glioblastoma U-87 cell line expresses wild type p53 [50,51]. Upregulation of p53 by sGCβ1 promoted p21 expression (Figure 6), and sGCβ1 activated p53 and p21 signaling associated with a notable G0 phase arrest (Figure 3a,b). p21 has been reported to play an antiapoptotic role [52,53]. Thus, increased expression of p21 may lead to G0 arrest and prevent apoptosis [54]. Importantly, evidence obtained in the present study supported a hypothesis that sGCβ1 is associated with nuclear p53 (Figure 5) and directly impacts G0 phase arrest. In contrast, extranuclear transcriptionally inactive p53, which acts in the cytosol and mitochondria to promote apoptosis, was less influenced by sGCβ1 (Figure 5b,c) [55,56]. Mutations of the *TP53* gene are the most frequent in lower grade glioma; however, the impact of sGCβ1 on the expression of mutant p53 is an important subject for further exploration.

The NO/sGC/cGMP signaling pathway undergoes variable changes in various tumors [6,44]. sGCα1 expression is elevated to a high level in prostate tumors, and the expression of sGCβ1 remains very low [57]. Similar to glioblastoma, sGCα1 is exclusively cytoplasmic in prostate cancer cells. Androgen upregulates direct binding of sGCα1 with cytosolic p53 in prostate cancer cells, diminishing p53 activity and exerting procarcinogenic effects [58]. Interestingly, similar to sGCβ1, sGCα1 induces cytoplasmic sequestration of p53 independently of NO signaling or guanylyl cyclase activity. Notably, sGCβ1 takes a distinct route to interact with the p53 pathway. sGCβ1 migrated into the nucleus and interacted with the *TP53* promoter to induce transcription-dependent mechanisms that resulted in a reduction in tumor aggressiveness (Figure 5).

The G0 phase comprises three states: quiescent, senescent, and differentiated. Each of these states can be entered from the G1 phase before the cells commit to the next round of the cell cycle [59,60]. Adult neuronal cells are fully differentiated and reside in the G0 phase. Neurons reside in this state as a part of their developmental program but not because of stochastic or limited nutrient supply [61]. Anaplasia is a state of the cells with poor cellular differentiation and is detected in most malignant neoplasms [62]. Malignant transformation includes a group of morphological changes, such as nuclear pleomorphism, altered nuclear–cytoplasmic ratio, presence of nucleoli, and high proliferation index. The concept of differentiation therapy has emerged from the fact that several therapeutic agents, such as hormones or cytokines, promote the ability of tumor cells to differentiate from an anaplastic status to irreversibly alter the phenotype of aggressive cancer cells [63]. The studies by our group previously reported low levels of sGCα1 and β1 expression in undifferentiated human and mouse embryonic stem cells. Embryonic stem cells regain sGC expression after entering the differentiation [64,65]. Many aspects related to tumorigenesis and organogenesis are similar, and many types of cancer (including brain tumors) contain cancer stem-like cells [66,67]. Our previous findings demonstrated that restoring sGC function inhibits the growth of glioma and normalizes cellular architecture [6], and the results of the present study suggest that a prodifferentiation mechanism is involved in sGC-targeted therapy and may be used as an approach alternative to various toxic treatments, such as chemotherapy and radiation.

Deregulation of the retinoblastoma (Rb) and p53 proteins has been pinpointed as an obligatory event in the majority of glioblastoma tumors [47]. CDK4 and CDK6 are the key components of the cell cycle machinery, driving the G1 to S phase transition via the phosphorylation and inactivation of the retinoblastoma protein [34]. The results of the present study demonstrated that sGCβ1 overexpression exerted insignificant effects on the phosphorylation levels of the retinoblastoma family proteins, such as pRb, p107, and p130 (Appendix A). In contrast, sGCβ1 overexpression markedly inhibited the expression of CDK6 and induced a trend of downregulation of CDK4 (Figure 6d–f). CDK6 plays the role of a transcriptional regulator, and this role is not shared by CDK4. Tumors with low levels of CDK6 have a higher-than-anticipated frequency of mutations of *TP53*. Furthermore, CDK6 kinase induces a complex transcriptional program to block p53 in cancer cells [68]. Thus, CDK6 acts at the interface of p53 and Rb by driving cell cycle progression and counteracts the p53-induced responses [69]. CDK6 expression levels inversely correlate with the status of the p53 pathway in mouse and human tumors [68]. Notably, CDK6 regulates the key genes involved in the survival, proliferation, and angiogenesis, including vascular endothelial growth factor A (*VEGFA*) [70,71].

The present study revealed a significant reduction in ITGA6 (Figure 6g–i) induced by sGCβ1. Integrins play a crucial role in tumor invasion and survival [72,73]. Comparison with other integrin isoforms indicates that ITGA6 is expressed at a high level in embryonic, hematopoietic, and neural stem cells [74]. Examination of ITGA6 expression in biopsy samples from glioblastoma patients indicated that ITGA6 is coexpressed with conventional glioblastoma stem-like cell markers and that ITGA6 is enriched in the perivascular niche [75]. Clinical relevance of ITGA6 was demonstrated using an in silico glioblastoma patient database to demonstrate that ITGA6 expression inversely correlates with survival (*p* = 0.0129) [75]. Thus, ITGA6 expression is elevated in glioblastoma stem-like cells, and this protein may be a target for therapeutic development. The data of the present study showed a marked expression of ITGA6 in U87 glioblastoma cells. Elevated expression of sGCβ1 resulted in de novo synthesis of p53 and may result in a reduction in ITGA6 in glioblastoma cells (Figure 5a and Figure 6g–i). The specific pathway involved in the effect of sGCβ1-induced blockade of ITGA6 is unknown. Potential clinical significance of our findings is due to the ability to target ITGA6 in glioblastoma cells via newly identified action of sGCβ1. In vivo studies demonstrated that ITGA6 blockade increases tumor latency and survival [75], suggesting that ITGA6 plays a role in tumor propagation. Moreover, previous reports showed that cancer stem-like cell-specific therapies may reduce tumor growth without an absolute termination of tumor growth [76]. sGCβ1 induces G0 arrest (Figure 3), and sGC expression inhibits glioma growth associated with normalized cellular architecture [6]. Thus, a new concept of glioblastoma treatment based on sGC is expected to integrate conventional and cancer stem-like cell-targeted therapeutic approaches.

## 5. Conclusions

The present study provides a rationale for the development of a novel concept of glioblastoma pathology and molecular therapeutic pathway. We revealed that the β1 subunit of sGC migrated into the nucleus and repressed the growth of human glioblastoma cells. Thus, transcriptional responses induced by sGCβ1 may cause the differentiation of cancer cells, which is important for a decrease in tumor aggressiveness. The sGCβ1 overexpression impacted signaling in glioblastoma multiforme, including the promotion of nuclear accumulation of p53, a marked reduction in CDK6, and significant inhibition of integrin α6. These anticancer targets of sGCβ1 have been validated by various clinical studies and by the development of therapeutic strategies for cancer treatment [77,78,79]. sGCβ1-based glioblastoma therapy is characterized by boosting normal endogenous signaling and promoting differentiation of glioma cells, which may transform the treatment by shifting the paradigm from the killing of cancer cells to differentiation-induced transformation of cancer cells. The present study reveals a new therapeutic approach for treatment of malignant cancer with lower general toxicity.

## Figures and Tables

**Figure 1 cancers-15-01567-f001:**
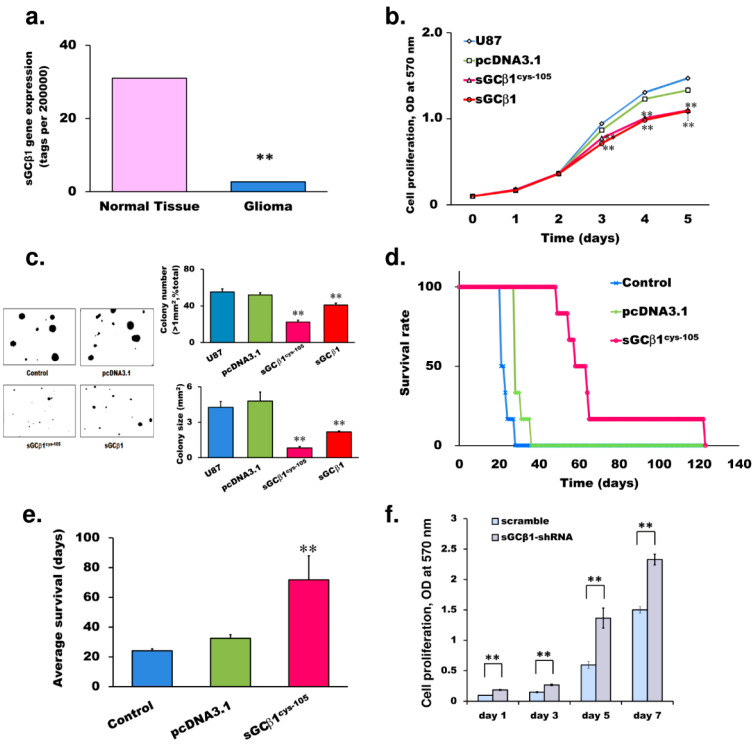
The effects of sGCβ1 overexpression or silencing on glioblastoma growth. (**a**) Serial analysis of gene expression (SAGA; GEO database GSE15309, *n* = 327). The results of analysis of the transcript levels in malignant and normal human tissues under various conditions indicated a statistically significant reduction in the levels of sGCβ1 transcript in human glioma specimens compared with that in normal brain tissue. (**b**) The proliferation of U87 cells transfected with a control vector or the vectors for overexpression of sGCβ1 or sGCβ1^Cys105^. Untransfected U87 cells were used as a control (*n* = 18 per group; the data were obtained by the MTT assay). (**c**) Colony formation assay of glioblastoma cells. U87 cells were transfected with a vector control or the vectors for overexpression of sGCβ1 or sGCβ1^Cys105^ (*n* = 6 wells per group). Average colony size and the numbers of the colonies larger than 1 mm^2^ are shown. (**d**,**e**) In vivo antitumor activity of sGCβ1 in athymic mice after intracerebral xenotransplantation of human glioblastoma cells transfected with a vector control or with the vector for overexpression of sGCβ1^Cys105^. Untransfected U87 cells were used as a control. The survival rate (**d**) and average survival time (**e**) are shown (*n* = 6 for each group). Log-rank test with Bonferroni correction was used to compare the survival curves in panel (**d**): control vs. vector *p* = 0.133 (not significant); control vs. sGCβ1^Cys105^
*p* = 0.005; and vector vs. sGCβ1^Cys105^
*p* = 0.009. (**f**) Proliferation assay of human neuroblastoma BE2 cells transfected with sGCβ1 shRNA or scrambled control on days 1, 3, 5, and 7 after plating. The data are the mean ± S.E.M.; ** *p* < 0.01 (vs. empty vector or control determined by one-way ANOVA with Bonferroni correction for panel (**e**).

**Figure 2 cancers-15-01567-f002:**
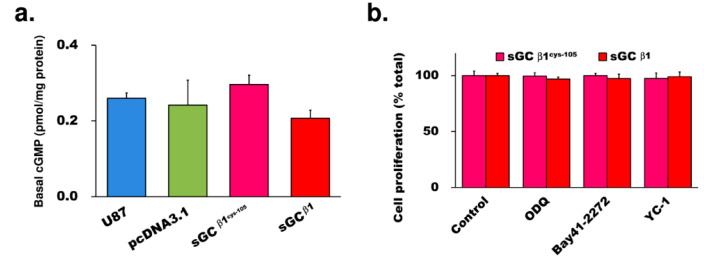
Assay of cGMP levels and the effects of sGC activators or inhibitors on sGCβ1-overexpressing cells. (**a**) cGMP levels were not significantly changed in U87 and pCDNA3.1-, sGCβ1^Cys105^-, and sGCβ1-transfected cells. (**b**) Proliferation assay of the cells with overexpression of sGCβ1^Cys105^ and sGCβ1 treated with ODQ (10 µM), Bay41-2272 (1 µM), or YC-1 (10 µM). Cell numbers were normalized to the numbers of untreated cells. There were no significant differences between the groups (*p* > 0.05).

**Figure 3 cancers-15-01567-f003:**
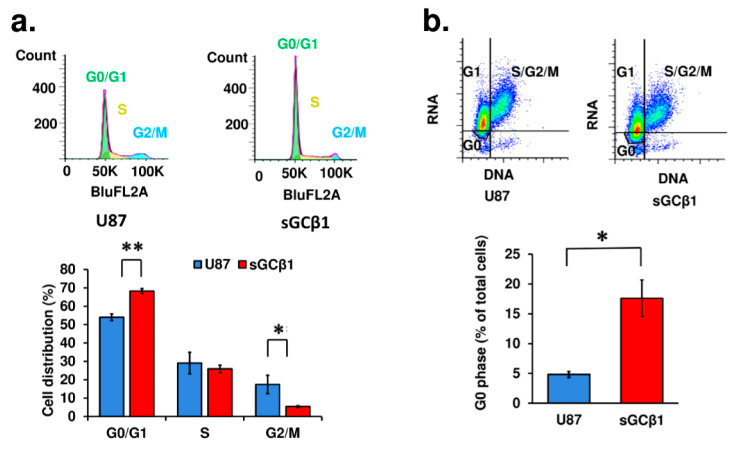
Cell cycle and proliferation/survival analysis of sGCβ1-overexpressing cells. (**a**) Cell cycle analysis of sGCβ1-overexpressing cells stained with propidium iodide by flow cytometry. G0/G1, S, and G2/M phases are indicated. (**b**) DNA and RNA content analysis of sGCβ1-overexpressing cells stained with 7AAD and Pyronin Y by flow cytometry. S/G2/M, G1, and G0 phase are indicated. The data were analyzed by using FlowJo software. The data are the mean ± S.E.M. *, *p* < 0.05; **, *p* < 0.01 (vs. empty vector or control).

**Figure 4 cancers-15-01567-f004:**
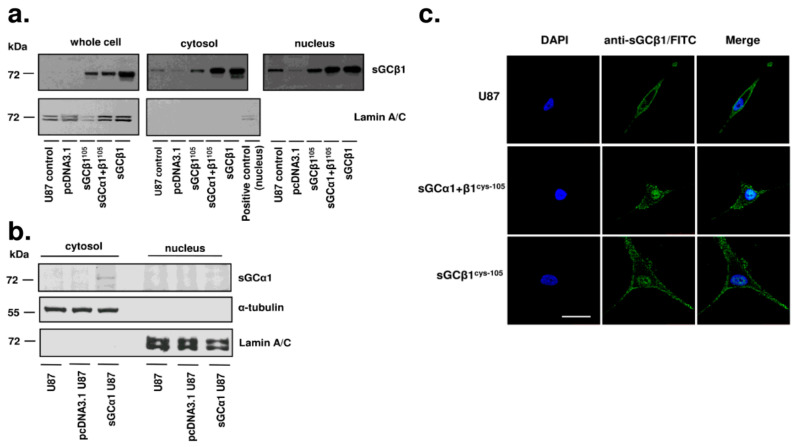
Subcellular localization of sGCβ1 and sGCα1 in human glioblastoma cells. (**a**) Immunoblotting analysis of U87 cells, U87 cells transfected with pcDNA3.1D control vector or the vectors for overexpression of sGCβ1^Cys105^, sGCα1 plus sGCβ1^Cys105^, and sGCβ1. An anti-sGCβ1 antibody was used to detect the distribution of sGCβ1 in the whole cell extract (left panel), cytosol (middle panel), and nucleus (right panel). Detection by an antilamin A/C antibody was used as a loading control. (**b**) Immunoblotting analysis of untransfected human glioblastoma U87 cells, U87 cells transfected with pcDNA3.1D control vector alone, and U87 cells with sGCα1 overexpression. An anti-sGCα1 antibody was used to detect the α1 subunit in the cytosol and nucleus. The lack of cross-contamination between the nucleus and cytoplasm was confirmed using the nuclear marker lamin A/C and cytoplasmic marker α-tubulin. (**c**) Immunostaining of sGCβ1 in control U87 cells (upper panel), an sGCα1 + β1^Cys105^–overexpressing stable clone (middle panel), and an sGCβ1^Cys105^–overexpressing stable clone (bottom panel). Blue represents the nuclei stained by DAPI, and green represents FITC staining of sGCβ1. The images were acquired by confocal microscopy at 600× magnification. Scale bars: 20 μm.

**Figure 5 cancers-15-01567-f005:**
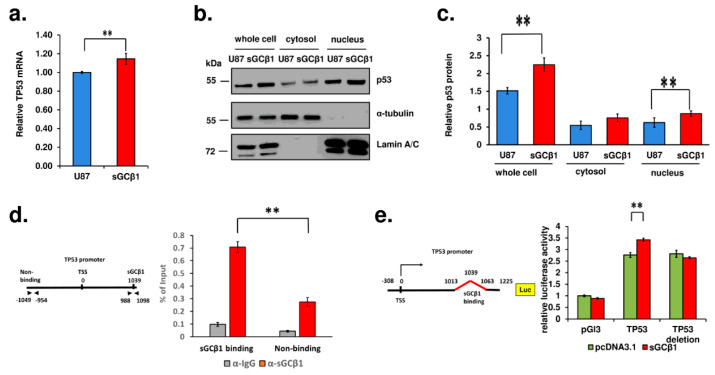
sGCβ1 interferes with p53 transcription. qRT-PCR ((**a**); *n* = 6) and Western blot ((**b**,**c**); *n* = 3) of p53 expression promoted by sGCβ1. A significant increase in the levels of p53 was detected after sGCβ1 overexpression ((**b**,**c**); *n* = 3). An anti-β-actin antibody was used as a loading control for whole cell extract. Original Western blot figure can be found in Appendix A, and a representative staining of actin is shown in Figure 6b,e,h). Antilamin A/C and anti-α-tubulin antibodies were used as the controls for the nuclear and cytoplasmic fractions, respectively. The data were normalized to the level of β-actin in the whole cell extract. (**d**) ChIP analysis of sGCβ1 binding to the *TP53* promoter. An anti-sGCβ1 antibody was used to immunoprecipitate sGCβ1 from a stable clone of sGCβ1-overexpressing U87 cells, and qRT-PCR was performed to amplify the DNA regions involved in sGCβ1 binding (1 kb downstream of the TSS) and a nonbinding region (1 kb upstream of the TSS). An IgG was used as a control. (**e**) Schematic representation of the cloned *TP53* promoter constructs. The 50 bp deletion mutation sites are shown. Dual luciferase assay was performed by using the *TP53* promoter constructs cloned into the pGL3 vector and the corresponding deletion mutants of these constructs without sGCβ1-binding sites. sGCβ1 overexpression plasmid was cotransfected into U87 cells with these plasmids. pcDNA3.1 was used as a control. The experiments were repeated 3 times, and the *p*-values were obtained by Welch- and Boferroni-corrected one-way ANOVA. The data are the mean ± S.E.M. **, *p* < 0.01.

**Figure 6 cancers-15-01567-f006:**
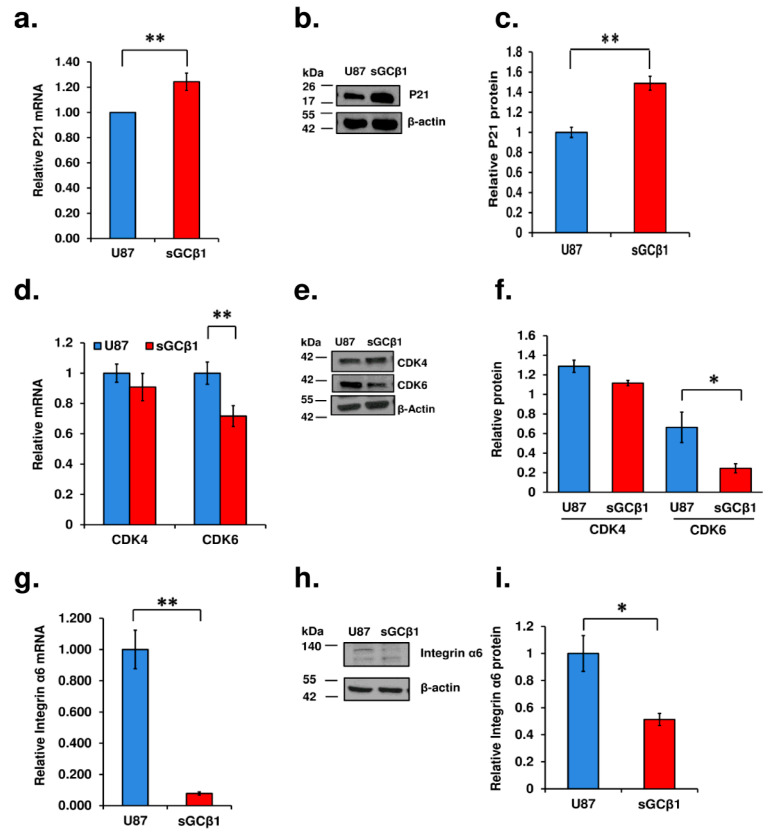
Effects of sGCβ1 overexpression on signaling in glioblastoma multiforme. qRT-PCR analysis of p21 (**a**), CDK4, CDK6 (**d**), and integrin α6 (**g**) expression in sGCβ1-overexpressing cells. Western blot analysis of p21 (**b**,**c**), Original Western blot figure can be found in Appendix A, CDK4, CDK6 (**e**,**f**), Original Western blot figure can be found in Appendix A, and integrin α6 (**h**,**i**) levels in sGCβ1-overexpressing cells in the whole cell extract, Original Western blot figure can be found in Appendix A. The data were normalized to the level of β-actin. pcDNA3.1D-transfected cells were used as a control. The data are the mean ± S.E.M. *, *p* < 0.05; **, *p* < 0.01 (vs. empty vector or control).

## Data Availability

The data presented in this study are available on request from the corresponding author.

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
