# Peer review of "Soluble Guanylate Cyclase β1 Subunit Represses Human Glioblastoma Growth"

_cancers, 2023, doi:10.3390/cancers15051567_

Round 1

Reviewer 1 Report

General comments to the Authors

Despite the positive reviews of the original versions of the manuscript, there are glaring weaknesses that significantly diminish enthusiasm for its potential clinical utility in antitumor effect of sGCβ1 on Malignant glioma progression. First, the study lacks requisite statistical power and replication to reliably validate the accuracy and reproducibility of its results and conclusions. Second, these studies uncover any detail mechanism that how sGCβ1 regulates nuclear accumulation integrin α6 singlainal axis connection, confers inhibition effect of Malignant glioma metastasis and radoresistance, and potentially regulates tumor cancer stemness. Lack of the examination of the effect of sGCβ1 family konckdown on normal Human brain Epithelial Cells, or brain normal stem cells. Third, the study is largely confirmatory of a previously published study by J Cell Physiol. 2020 Feb;235(2):1358-1365.; PLoS One. 2015 Apr 30;10(4):e0125518., FASEB J. 2016 Sep;30(9):3171-80., Nitric Oxide. 2010 Jan 1;22(1):43-50.; therefore lacks significant novelty.

Reviewer 2 Report

The study by Xiao et al. aimed to analyze the cellular and molecular mechanisms underlying the tumoristatic effects of sGCβ1 (soluble guanylyl cyclase β1) in glioblastoma cells as disclosed previously by the same group. To this end, Xiao et al. stably transfected sGC low-expressing human U87 glioblastoma with full length sGCβ1 or with the constitutively active sGCβ1Cys105 mutant alone or together with sGCα1. In addition, they inhibited and activated sGC activity pharmacologically in sGCβ1- and sGCβ1Cys105-transfected U87 cells. Moreover, the authors transiently downregulated in sGC high-expressing neuroblastoma BE2 cells sGCβ1 by RNA interference. Functionally, they analyzed clonogenic survival (colony formation assay), cell proliferation (MTT assay), cell cycle progression and apoptosis (flow cytometry), cGMP formation (ELISA), subcellular sGCβ1 and sGCα1 distribution (confocal microsopy, immunoblotting), as well as signaling and gene expression (Immunoblotting, qRT-PCR, chromatin immunoprecipitation, luciferase promoter activity assay). Moreover, they orthotopically xenotransplanted control vector- or sGCβ1Cys105-tranfected U87 cells in nu/nu athymic mice and determined the time until mice became moribund.

As a result, Xiao et al. observed that sGCβ1-overexpression in U87 glioma cells was associated with a decrease in proliferation rate and clonogenic survival and an entry in G0 resting phase of cell cycle. Consistently, sGCβ1 downregulation in neuroblastoma cells increased proliferation rate. In addition, mice with sGCβ1Cys105transfected U87 glioma survived longer than mice with control-transfected or parental U87 tumors. Mechanistically, the authors demonstrated that the tumoristatic effect of sGCβ1 overexpression was not dependent on enzyme activity and cGMP formation. Instead, Xiao et al. demonstrated that sGCβ1 in contrast to sGCα1 was located in the nucleus where it induced transcription of the TP53 gene as evident from luciferase reporter assays comparing the effect of sGCβ1 overexpression on full length and putative sGCβ1 binding site-deleted TP53 promoter. sGCβ1 overexpression induced in U87 cells nuclear accumulation of p53 protein that was paralleled by upregulation of p21 and downregulation of CDK6 and integrin-α6 on mRNA and protein levels. From their data, the authors conclude that in sGCβ1 expression-restored glioblastoma cells, transcriptional activity of nuclearly translocated sGCβ1 “differentiates” the tumor cells in a proliferation-locked less malignant phenotype by upregulation of p53/p21 expression and paralleled downregulation of CDK6 and integrin-α6.

General

This is a clinically relevant meticulously performed and accurately presented in vitro and mouse study with clear graphics and conclusions that are largely supported by the data. By its very nature, studying a single, p53 wild type glioma cell model (U87) limits the generalizability of the findings (especially when considering p53-mutated gliomas). Nevertheless, the disclosed cellular mechanisms of the tumoristatic action of experimental sGCβ1 re-expression possibly via upregulation of p53/p21 are novel findings and may be harnessed for strategies of “differentiation” therapies in glioblastoma in the future.

Specific comments

-statistics: error probabilities of the multiple pairwise comparisons (e.g., Figs. 1b, c, e, 2a, 6f) should be Bonferroni-corrected for the number of pairwise comparisons applied. Mouse data in Fig. 1d should be analyzed by log-rank test and calculated error probabilities Bonferroni corrected for 3 pairwise comparisons. Samples with differing SDs (maybe in Fig. 2b, bottom or Fig. 1e and 5a) should be compared with Welch-corrected t-test and Welch-corrected ANOVA.

-Fig. 5: albeit being significantly different, the sGCβ1-overexpression-associated changes in TP53 mRNA abundance (fig. 5a) are much lower than the increase in p53 protein abundance (fig. 5c, 1st and 2nd bar) suggesting that sGCβ1-mediated p53 protein stabilization might also play a role in the observed phenomenon (for fig. 5b, please give ß-actin bands, used to normalize the p53 protein abundance in fig. 5c, otherwise it is not possible to match the information of 5b and 5c. Please, also provide the other 2 blots including ß-actin bands by Suppl. files). Altogether, the observed sGCβ1-induced changes in mRNA abundance (about 1.1-fold) seem to me very low. Please, discuss their potential functional relevance.

-line 442 “The p53 pathway is inactivated in almost all glioblastomas.” According to the TCGA, TP53 mutations were detected in only 33% of glioblastoma resection specimens. Please, refine this statement.

-Is anything known on the mechanisms underlying the marked downregulation of sGCβ1 in glioblastoma?

Round 2

Reviewer 1 Report

The authors have made conscientious effort to improve the quality of their manuscript by addressing the critiques of the reviewers in a point-by-point manner and properly adhering to the suggestions given by the authors.